# OFFLINELIGHT: AN OFFLINE REINFORCEMENT LEARNING MODEL FOR TRAFFIC SIGNAL CONTROL

## ABSTRACT

Reinforcement learning (RL) is gaining popularity in addressing the traffic signal control (TSC) problem. Yet, the trial and error training with environmental interactions for traditional RL-based methods is costly and time-consuming. Additionally, it is challenging to directly deploy a completely pre-trained RL model for all types of intersections. Inspired by recent advances in decision-making systems from offline RL, we propose a general offline actor-critic framework (Offline-AC) that considers policy and value constraints, and an adaptive decision-making model named OfflineLight based on Offline-AC. Offline-AC is further proved general and suitable for developing new offline RL algorithms. Moreover, we collect, organize and release the first offline dataset for TSC (TSC-OID), which is generated from the state-of-the-art (SOTA) RL models that interact with a traffic simulation environment based on multiple datasets of real-world road intersections and traffic flow. Through numerical experiments on real-world datasets, we demonstrate that: (1) Offline RL can build a high-performance RL model without online interactions with the traffic environment; (2) OfflineLight matches or achieves SOTA among recent RL methods; and (3) OfflineLight shows comprehensive generalization performance after completing training on only 20% of the TSC-OID dataset. The relevant dataset and code are available at anonymous URL[1].

## 1 INTRODUCTION

Traffic signal control (TSC) is essential for enhancing road safety and alleviating traffic congestion by coordinating multi-directional traffic flows at intersections. Traditionally, TSC is a decentralized system to control the traffic signal according to predefined rules. Back in 1950s, Webster developed a fixed-time traffic signal cycle method for TSC, which is still widely applied in various traffic networks considering its practicality and simplicity Webster (1958). After that, SCOOT Hunt et al. (1982) and SCATS Lowrie (1990) were designed to change traffic signals according to the traffic conditions, relying on predetermined traffic signal plans by experts. However, it is becoming much more challenging to develop efficient and effective TSC solutions owing to various factors, such as increasing number of vehicles, varying weather conditions, and heterogeneous driving behaviours Rios-Torres & Malikopoulos (2017).

Reinforcement learning (RL) provides a formalism for decision making, which can actively learn in unstructured environments Sutton & Barto (2018). Recently, RL has been gaining popularity in addressing the TSC problem Yau et al. (2017); Wu et al. (2022). The RL-based approaches can learn from the traffic environment's feedback and adjust the traffic signal policies accordingly. Thus, they generate and execute different actions by observing the traffic conditions, such as lane capacity Wei et al. (2019b) and traffic movement pressure Chen et al. (2020). Deep learning LeCun Y. & G (2015) further facilitated RL-based approaches for TSC Wei et al. (2019a); Zheng et al. (2019); Oroojlooy et al. (2020); Zhao et al. (2021); Wu et al. (2022), which has demonstrated much better performance than those conventional methods. While the RL-based methods have tremendous success, their deployment in practice is rare due to the overhead of their extensive online training process and the limited capability of generalization for various intersections.

The recent progress of offline RL shows that it is possible to train the RL model for satisfactory policies with previously collected data without online environmental interactions Levine et al. (2020).

---

[1] https://anonymous.4open.science/r/OfflineLight-6665/README.md

Offline RL aims to derive and stitch a policy from the behaviors extracted from the historical dataset for better decision-making. However, offline RL suffers from data distributional shift issues, leading to overestimating policy values, while online RL methods can rectify it with corresponding exploration strategies Lee et al. (2021). Thus, errors in sampling and function approximation are more severe in offline RL. Now, out-of-distribution (OOD) issue can be partially addressed by constraining the policies or values on traditional RL methods Levine et al. (2020). This indicates that the offline RL has excellent potential for creating sophisticated decision-making systems from massive datasets Fu et al. (2020) by addressing the OOD issue Kumar et al. (2020; 2019). To the best of our knowledge, offline RL has not yet been applied to real-life data of road networks and traffic flow for TSC.

In general, the challenges in online and offline RL-based methods for TSC can be summarized as the following points. Firstly, the trial and error training with environmental interactions for traditional RL-based methods is costly and time-consuming. Secondly, it is challenging to directly deploy a pre-trained RL model to all types of intersections. Thirdly, while recent offline RL-based methods can partially address the OOD issue, they are rarely used in TSC. Lastly, there is no open-source offline intersection dataset for TSC.

To address these challenges, we have considered optimizing offline RL for adaptive and efficient TSC. Hence, the main contributions of this paper are summarized as follows. We introduce a general offline actor-critic framework (Offline-AC) and demonstrate its versatility and suitability for developing new offline RL algorithms. Furthermore, we develop an adaptive decision-making model, namely OfflineLight, based on the Offline-AC for TSC. Ultimately, we significantly improve the generalization capability under a variety of traffic conditions. Additionally, we create, compile, and release the first-ever offline dataset for TSC (TSC-OID), generated from cutting-edge RL models interacting with a traffic simulation environment.

## 2 RELATED WORKS

### 2.1 CONVENTIONAL APPROACHES FOR TSC

Conventional TSC approaches can be divided into two main categories: predefined control Webster (1958) and feedback control Lowrie (1990). Predefined control approaches can change signal phases according to rules predefined for signal plans. Webster proposed a classic fixed-time control method in 1958, consisting of a pre-timed and fixed cycle length for phase sequence Webster (1958). The predefined control methods tended to mechanically control the traffic signal so they could not adapt to the changing vehicle flow. In contrast, feedback signal control methods change traffic signals based on traffic conditions. The renowned traffic control systems such as SCOOTS Hunt et al. (1982) and SCATS Lowrie (1990) manually set thresholds to determine the change of traffic signal after obtaining the traffic conditions by sensors, which are still widely deployed in real-life TSC.

### 2.2 RL-RELATED METHODS FOR TSC

Reinforcement learning (RL) is gaining popularity in addressing the traditional TSC problem, as it learns from trial-and-error interactions with traffic environments. Essentially, an RL algorithm generates and implements various TSC policies based on feedback from the simulators of traffic environment, rather than making assumptions on traffic models. Here, high-fidelity simulators must be manually created for the transition from simulation to real-world implementation. Different application scenarios, such as single intersection control Li Li & Feiyue (2016) and multi-intersection control Rasheed et al. (2020), have been developed using RL-related methods. In 2019, FRAP model Zheng et al. (2019) developed a particular neural network structure to embed phase features and extract phase competition relations. Furthermore, it leads to better RL-based solutions for TSC, such as PressLight Wei et al. (2019a) and CoLight Wei et al. (2019b). In 2020, AttendLight Zhang et al. (2022a) was proposed, adopting the attention network to handle the various topologies of inter-sections and achieving better generalization performance. PRGLight Zhao et al. (2021) dynamically adjusted the phase duration according to the real-time traffic conditions. RL-based methods such as CoLight Wei et al. (2019b) and Advanced-CoLight Zhang et al. (2022b) are still SOTA while recent TSC approaches Levin et al. (2020); Wu et al. (2021); Zhang et al. (2022b) have also shown rather competitive performances.

### 2.3 OFFLINE RL

Offline RL, as opposed to traditional (online and off-policy) RL approaches, changes the focus of training mode and learns from trial-and-error offline data without environmental interactions for a proper policy Levine et al. (2020). Recent progress in offline RL methods has demonstrated the advantages of data-driven learning of policies. According to recent works Levine et al. (2020), OOD issue can only be partially addressed by constraining the learning policies Fujimoto et al. (2018a); Kumar et al. (2019) or bounding overestimated policy values Kumar et al. (2020). Batch-constrained reinforcement learning (BCQ) Fujimoto et al. (2018a), introduced off-policy algorithms, restricting the action space to force the agent towards behaving close to on-policy. TD3+BC Fujimoto & Gu (2021) was a simple approach to offline RL where only two changes were made to TD3 Fujimoto et al. (2018b): (1) a weighted behavior cloning loss was added to the policy update and (2) the states were normalized. BEAR Kumar et al. (2019) was able to learn robustly from different off-policy distributions, including random and sub-optimal demonstrations, on a range of continuous control tasks. Conservative Q-learning (CQL) Kumar et al. (2020), aimed to address these limitations by learning a conservative Q-function so that the expected value of a policy under this Q-function could reach lower bounds than its true values. CQL has achieved the excellent performance in current offline RL tasks, such as robot control and Atari games, but has not been applied to TSC yet.

## 3 PRELIMINARY

### 3.1 DEFINITIONS OF TSC

We utilize a four-way crossroad as an example of a typical intersection with four traffic signal phases to illustrate pertinent concepts, as shown in Figure 1.

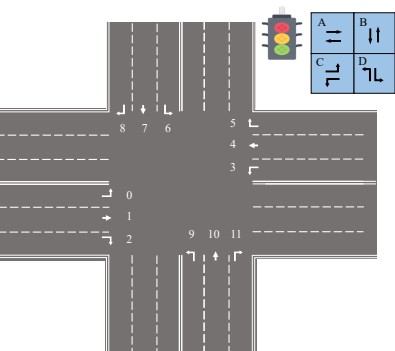

Figure 1: Illustration of a 4-approach intersection with four traffic signal phases. There are twelve traffic movements (1 to 12) and four traffic signal phases ($A$ to $D$).

Traffic network: Each traffic network consists of several intersections $(I_1, ..., I_N)$ connected by roads, where $N$ is the number of intersections.

Traffic movement: Vehicles crossing an intersection from one entering lane $i$ to an exiting lane $o$ are represented by a traffic movement (TM), which is denoted as $(i, o)$.

Traffic signal phase: A pair of single signals (TM pairs without conflicts) combined to form a signal phase is indicated by $s = \{(i_1, o_1), (i_2, o_2)\}$, where in $(i_1, o_1)$ and $(i_2, o_2)$ are two TMs with entering lane $i_1, i_2$ and exiting lane $o_1, o_2$.

### 3.2 PRELIMINARY OF RL

MDP: The Markov decision process (MDP) is defined as $M = (S, A, P, r, \gamma)$, where $S$ denotes a set of states, $A$ is a set of actions, $P$ defines a transition probability, $r$ is a reward function, and $\gamma$ ($\gamma \in [0, 1]$) is a discount factor. The RL algorithm usually learns a policy $\pi$ online for MDP to determine the best action $a$ ($a \in A$) considering the given state $s$ ($s \in S$). The objective of RL is to

maximize the expected return $G_t$ at time $t$:

$$G_t = \sum_{t=0}^{+\infty} \gamma^n r_t \tag{1}$$

Offline RL: A data-driven RL, whose objective is also Equation (1). The offline agent collects historical transitions and learns policies underlying the MDP from a static dataset of transitions $\mathcal{D} = \{(s_t^i, a_t^i, s_{t+1}^i, r_t^i)\}$, without interacting with the environment. We define $\pi_\beta$ as the distribution over states and actions in dataset $\mathcal{D}$. The state-action transitions $(s, a)$ are sampled from the distribution $s \sim d^{\pi_\beta}(s)$, and the actions are sampled according to $a \sim \pi_\beta(a|s)$.

### 3.3 PROBLEM DESCRIPTION

We aim to design a decentralized RL-based TSC method with focus on the optimization of independent intersection. We consider a multi-intersection TSC optimization problem. Most existing RL-based TSC methods still need to determine which states and rewards are the best choices. We use a variety of states and rewards from the dataset to enhance the model's generalization. (1) States: The state of the system at time step $t$ is represented by $s_t^i$, which is the partial observed information available to intersection. The traffic states include the number of vehicles (NV), the number of vehicles under segmented roads (NV-segments), the queue length (QL), the efficient traffic movement pressure (EP), effective running vehicles (ERV), and the traffic movement pressure (TMP). (2) Action: At each time step, intersection $Inter_i$ selects an action $a_t^i$ from its action space, which corresponds to the signal control decision made by the traffic signal controller. (3) Reward: The immediate reward $r_t^i$ is obtained after taking action $a_t^i$ in state $s_t^i$ and transitioning to state $s_{t+1}^i$. The goal is to maximize the cumulative expected reward over trajectories $\mathcal{D} = \{(s_t^i, a_t^i, s_{t+1}^i, r_t^i)\}$ for an intersection, which is defined as:

$$\mathbb{E}_{(s,a,s') \sim \mathcal{D}} [\sum_{t=0}^{+\infty} \gamma^t r_t] \tag{2}$$

In this paper, our objection is to find the optimal policy $\pi^*$ that maximizes the expected reward over the real trajectories $D'$ based on the historical trajectories $D$: $\pi^* = \max_{\pi^*} \mathbb{E}_{(s,a,s') \sim (\mathcal{D}'|D)}$. Here $D'$ and $D$ have 'out of data distribution' issues.

## 4 METHODS

### 4.1 METHOD FOR OFFLINE DATASETS GENERATION

Offline TSC datasets are essential for diversity and transferability for offline TSC algorithms. To generate the first offline dataset for TSC (TSC-OID), we employ two groups of real-world datasets (JiNan and HangZhou) of road networks and traffic flow. Then, we expand on the use of varied states and rewards in the dataset to enhance generalization. Our offline dataset is formed similarly to off-policy RL data buffers, generated through diverse RL methods interacting with CityFlow traffic simulator Zhang et al. (2019), encompassing multiple frameworks, states, and rewards. These RL methods correspond to different frameworks, traffic states, and rewards. See Appendix B.3 for details.

### 4.2 OFFLINE ACTOR-CRITIC FRAMEWORK

We first design the offline Actor-Critic framework (offline-AC), leveraging action behavior regularization and value pessimism for the Actor and Critic, respectively. As shown in Figure 2 (b), the Actor periodically updates its policy $\pi = \pi'$, according to the offline dataset $\mathcal{D}$ and Critic's value signal. While for the online Actor-Critic framework, the Actor updates its policy online, according to the Critic's signal and trajectory of interacting with the environment (as shown in Figure 2 (a)), which drives all learning progresses in both Actor and Critic.

For offline-AC, the Actor derives an approximate gradient by using the state distribution of the behavior policy $d^{\pi_\beta}(s)$, updating its policy $\pi$ online according to the Critic's value signal $Q^\pi(s, a)$. Meanwhile, the Critic repeatedly adjusts its Q-value $Q^\pi(s, a)$ only based on offline dataset $\mathcal{D}$. To

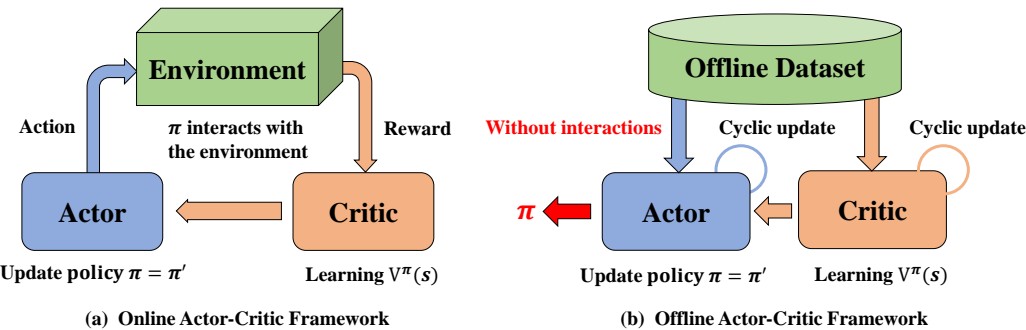

Figure 2: The illustration of Actor-Critic architecture, consists of online and offline Actor-Critic frameworks.

address the data distribution shift issues, the Actor is needed to be designed for generating similar policies with the dataset, and the Critic is applied to address overestimated values of the Critic.

### 4.2.1 OFFLINE ACTOR

The offline Actor is designed to learn a policy close to the behavior policy by constraining the action policy. The objective to the offline policy is:

$$J(\theta) = \frac{1}{N} \sum_{n=1}^{N} \sum_{t=1}^{T_n} Q^\pi(s,a)|_{a=\pi_\theta(s)} - \alpha Dis(\pi_\theta, \pi_\beta) \tag{3}$$

where $Q^\pi(s,a)$ is the value signal from the Critic, actions $\pi_\beta$ are sampled from offline data distribution and $\pi_\theta$ are actions we need to learn. We add the policy distance function between $\pi_\theta$ and the behavior policy $\pi_\beta$ as $Dis(\pi_\theta, \pi_\beta)$. The choice of this distance function can be flexibly implemented. To maximize the objective function, the policy gradient is $\bigtriangledown J(\theta)$.

### 4.2.2 OFFLINE CRITIC

The offline Critic is designed to address the overestimation of values between the offline dataset and the learned policy. Thus, the Critic needs to design a Q-function $Q^\pi(s_t, a_t)$ that could achieve lower-bounds of the policy $\pi$ evaluation: $Q^\pi(s,a) \leq Q(s,a)$, where $Q(s,a)$ is the true Q value, $Q(s,a) = \sum_{t'=t}^{+\infty} \gamma^{t'-t} r_{s'_t, a'_t}^n$. The Critic augments the standard Bellman error objective with a flexible and pessimistic Q-value design.

Note: We have designed the offline-AC framework as an abstract model for brevity, where the specific Actor and Critic algorithm needs to be further implemented in real practice. Additionally, the offline-AC also aims to enable flexibility for future algorithmic innovations for TSC.

### 4.3 OFFLINELIGHT

We develop a specific OfflineLight model for TSC implemented by the offline-AC framework based on the deep deterministic policy gradients (DDPG) Lillicrap et al. (2016) (as shown in Figure 3).The implementation of OfflineLight is built upon two proven RL methods: TD3 Fujimoto et al. (2018b) and CQL Kumar et al. (2020). Hence, we implement the Actor and Critic based on key components of the TD3 and CQL methods with specific consideration, respectively.

### 4.3.1 ACTOR IMPLEMENTATION

We have designed two policy networks in the Actor for TSC policy generation : $\pi_\theta$ and $\pi_{\theta'}$, where $\pi_{\theta'}$ is the target policy network by constraining the policy.

The target $\pi_{\theta'}$ with parameter $\theta'$, is selecting action with policy smooth regularization (PSR) from TD3 method Fujimoto et al. (2018b), while the $\pi_\theta$ without PSR is used for updating better action.

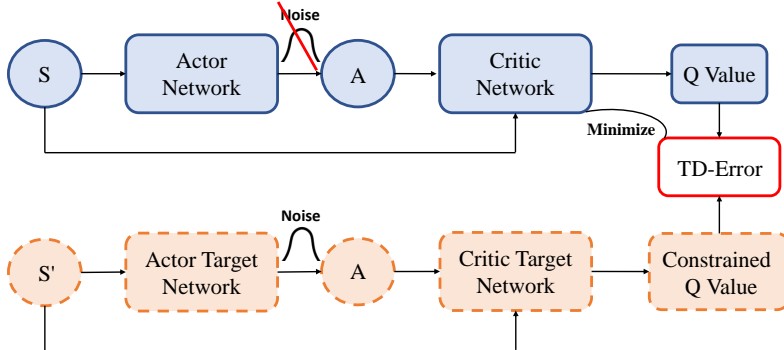

Figure 3: The network design of OfflineLight, which implements the Offline-AC based on DDPG.

The PSR contains two steps in an action: clips and selects an action with exploration noise: (1) clip exploration noise: $\epsilon \sim clip(\mathcal{N}(0, \sigma), -c, c)$ (2) select an action: $a' \sim (\pi_{\theta'} + \epsilon)$.

In addition, we apply Kullback–Leibler (KL) divergence $KL(\pi_\theta, \pi_\beta)$ to implement the distance function $Dis()$ for regularisation. Thus, the Actor is also designed to constrain the policy distribution between $\pi_\theta$ and the behavior policy $\pi_\beta$. The specific objective function of the Actor is :

$$J(\theta) = \frac{1}{N} \sum_{n=1}^{N} \sum_{t=1}^{T_n} Q^\pi(s, a)|_{a=(\pi_\theta)} - \alpha KL(\pi_\theta, \pi_\beta) \tag{4}$$

where $Q_\phi^\pi(s, a)$ is a valuation of action $a$ from Critic with parameter $\phi$.

In general, the Actor implementation applies PSR to overfitting to narrow peaks in the value estimate for deterministic policies by adding a small amount of random noise to the target policy $\pi_{\theta'}$. Moreover, it uses KL divergence to penalize divergence between learned policy $\pi_\theta$ and offline data policy $\pi_\beta$.

### 4.3.2 CRITIC IMPLEMENTATION

Also, we designed two Q networks in the Critic for more accurate Q-value: : $Q_\phi$ and $Q_{\phi'}$, where $Q_\phi$ is the Q network for value of pessimism while $Q_{\phi'}$ is target Q network. The Q network is used as a simple Q-value regularizer Kumar et al. (2020), which is constrained Q-value: $\hat{Q}^{k+1}$ and updated by:

$$\hat{Q}^{k+1} \leftarrow \arg\min \alpha(\mathbb{E}_{s \sim \mathcal{D}, a \sim (\pi_{\theta'} + \epsilon)}[Q'(s, a)] -$$
$$\mathbb{E}_{s,a \sim \mathcal{D}}[Q'(s, a)]) + \frac{1}{2} \mathbb{E}_{s,a \sim \mathcal{D}}[Q'(s, a) - \hat{\beta}^\pi \hat{Q}^k(s, a)]^2) \tag{5}$$

where $\hat{Q}$ is lower-bounds of the target $Q'(s, a)$, $\mathbb{E}_{s \sim \mathcal{D}, a \sim (\pi_{\theta'} + \epsilon)}[Q'(s, a)]$ is to minimize overestimated Q-values for the target policy with PSR, $\mathbb{E}_{s,a \sim \mathcal{D}}[Q'(s, a)])$ is to maximize offline Q-values on dataset $\mathcal{D}$. And $\hat{\beta}^\pi$ is empirical Bellman operator in dataset $\mathcal{D}$ with transition probability $P(s'|s, a)$: $\hat{\beta}^\pi \hat{Q}^k(s, a) = r(s, a) + \gamma E_{s' \sim P(s'|s,a)}[\max_{a'} \hat{Q}(s, a)]$. Thus, $\mathbb{E}_{s,a,s' \sim \mathcal{D}}[Q(s, a) - \hat{\beta}^\pi \hat{Q}^k(s, a)]^2$ in equation (5) is a standard Bellman error. In addition, it has been proven that CQL can prevent overestimation by learning lower-bound Q-values: $\hat{Q}(s, a) <= Q'(s, a)$. We detail the critic convergence proof in Appendix D.

### 4.3.3 OFFLINELIGHT ALGORITHM

The specific settings of hyperparameters and network structures for OfflineLight are detailed in Appendix B.5 and Appendix B.4.

We summarize the OfflineLight algorithm as Algorithm 1. All trajectories' tuple $(s, a, r, s')$ are from the offline dataset TSC-OID, defined as $\mathcal{D}$.

---

**Algorithm 1** OfflineLight

---

**Input**: Offline dataset $\mathcal{D}$

**Initialize**: Actor network $\pi_\theta$, Critic network $\hat{Q}_\varphi$.

**Initialize**: Target Actor network $\pi_{\theta'}$, Targe Critic network $Q'_{\varphi'}$ : $\theta' \leftarrow \theta$, $\phi' \leftarrow \phi$

---

 1: **for** $t = 1$ to $T$ **do**
 2:     Sample mini-batch of $N$ transitions $(s, a, r, s')$ from $\mathcal{D}$
 3:     $a' \leftarrow \pi_{\theta'}(s') + \epsilon, \epsilon \sim clip(\mathcal{N}(0, \sigma), -c, c)$
 4:     Update Critic $\hat{Q}_\varphi$ by equation (5)
 5:     **if** t mod d **then**
 6:         Update $\theta$ by policy gradient:
             $\bigtriangledown J(\theta) = \frac{1}{N} \sum \bigtriangledown_a Q^\pi(s,a)|_{a=(\pi_\theta)} - \alpha \bigtriangledown_\theta KL(\pi_\theta, \pi_\beta)$
 7:         Update target policy network:
             $\theta' \leftarrow \tau\theta + (1 - \tau)\theta'$
 8:         Update target Q network:
             $\phi' \leftarrow \tau\phi + (1 - \tau)\phi'$
 9:     **end if**
10: **end for**

---

## 5   Discussion

**Why is the Offline-AC the abstract method?**

We design the abstract Offline-AC, considering both action behavior regularization and pessimistic value-function for offline RL. If the hyperparameter $\alpha$ in equation (3) is set to 0, it is a typical Q-learning network; when the Q value is constrained in equation (5), it is a recent CQL algorithm Kumar et al. (2020). If a neural network implements the policy without limiting the Q value, it can be expressed as a BEAR method Kumar et al. (2019) based on policy constraints. Thus, Offline-AC has interfaces to facilitate algorithmic innovations for implementing a specific offline RL. Moreover, Offline-AC is a general framework for offline RL, in other words, it is not limited to TSC tasks.

**What is the difference between OfflineLight and DDPG-based methods?** Compared to DDPG-based methods, such as DDPGLillicrap et al. (2016) and TD3Fujimoto et al. (2018b): (1) DDPG-based methods are designed for online RL, while OfflineLight is for offline RL. (2) Although the Actor of OfflineLight leverages PSR and the actor delayed the from TD3, the Offline-Light implements the distance function $Dis()$ with $KL$ and applies CQL for the target Critic network.

**Why do we use offline datasets, which are generated from the simulator, instead of online learning on the simulator?**

We aim to verify the performance of offline RL models in offline dataset, conveniently and reasonably comparing the online RL approaches' results from the simulator. To our best knowledge, there is no offline dataset for TSC yet. In general, it is a better way to develop offline RL in offline dataset for furtherly bridging real-world environments.

**What's the goodness of replacing the distance function with KL divergence?**

We set the policy distance function between $\pi_\theta$ and the behavior policy $\pi_\beta$ as $Dis(\pi_\theta, \pi_\beta)$. The choice of this distance function can be flexibly implemented. In OfflineLight, we apply KL divergence $KL(\pi_\theta, \pi_\beta)$ to implement the distance function $Dis()$ for regularisation. Additionally, JS divergence, also known as JS distance, is a variant of KL divergence. we also can use JS divergence to implement the distance function $Dis()$ As shown in Appendix C.2, the performance of JS-div is a little bit degraded compared to KL-div, which can prove that using KL-div is more effective than JS for implementing our offline-AC framework.

**What are the main contributions in this paper?**

By combining policy and value-constrained perspectives, we propose an abstract offline RL framework (offline AC), which to our best knowledge, is the first of its kind and can promote the implementation

of multiple offline RL algorithms rather than just one specific method. Additionally, we further implement the framework and achieve SOTA results in TSC. Our method provides new insights into the OOD problem of offline RL and releases offline TSC datasets for further research.

# 6    EXPERIMENTS

We first train the offline RL methods on $JiNan^1$ (about 20% of all the TSC-OID datasets) only and then all the TSC-OID datasets separately. Later, we conduct numerical experiments for online and offline RL approaches to obtain the TSC results for a fair comparison. The training and experimental process are on an open-source simulator CityFlow Zhang et al. (2019), which has been widely used by multiple RL-based methods Wei et al. (2019a;b); Chen et al. (2020).

## 6.1    DATESETS

The datasets consist of two parts, one is offline interaction data (TSC-OID) and the other is the traffic flow and road networks in the real world. We use seven real-world traffic datasets (JiNan, HangZhou, NewYork) Zheng et al. (2019); Wei et al. (2019b) for comparison. We detail the datasets in Appendix B.1. However, the offline RL methods could be applied to the New York dataset even if they have not been used to train it.

## 6.2    EVALUATION METRICS

We use the average travel time (ATT) to evaluate the performance of different approaches for TSC, which is widely and frequently adopted in the literature  Wei et al. (2019c). The ATT computes the average travel time for all the vehicles observed between entering and leaving the traffic network (in seconds), where $ATT = \frac{1}{I} \sum_{i=1}^{I} t_{v_i}$, $I$ is the number of vehicles and $t_{v_i}$ is the time duration between entering and leaving the traffic network for one vehicle.

## 6.3    METHODS FOR COMPARISON

We compare our method with the baseline methods, including traditional online RL and recent offline approaches (detailed in Appendix B.2).

**Traditional RL Methods:** The traditional RL Methods consist of the approaches for generating TSC-OID, including FRAP Zheng et al. (2019), AttendLight Oroojlooy et al. (2020), CoLight Wei et al. (2019b), AttentionLight Zhang et al. (2022a), Efficient-MPLight Wu et al. (2021), Advanced-CoLight Zhang et al. (2022b)). The other RL methods such as PRGLight Zhao et al. (2021) and AttendLight Oroojlooy et al. (2020) are not employed to generate the TSC-OID, for a fair comparison.

**Behavior Cloning (BC) :** We employ Behavior Cloning (BC) Torabi et al. (2018) to provide the performance of a pure imitative method.

**Offline RL Methods:** We apply the latest offline RL method CQL Kumar et al. (2020), TD3+BC Fujimoto & Gu (2021), BEAR Kumar et al. (2019), and Combo Yu et al. (2021) for comparison.

## 6.4    OVERALL RESULTS

We conduct numerical experiments on the real-world datasets for online and offline RL approaches to obtain TSC results. For a clear comparison, the methods are divided into two parts: online-RL trained on real-world datasets (road network topology and traffic flow), offline methods trained on TSC-OID. Table 1 reports our experimental results under JiNan, HangZhou, and New York real-world datasets with respect to the evaluation metrics, i.e., ATT. Because the New York dataset has not been used for TSC-OID generation, it reasonably supports our results. Thus, the variances of results from the New York dataset are zero. The bold numbers represent the optimal results in online and offline RL, respectively.

From the overall results, we have the following findings. (1) With the properly-designed framework of Offline-AC, OfflineLight can be close to the online-RL SOTA method Advanced-CoLight and outperform most of the online RL methods. (2) Our proposed OfflineLight consistently shows more

| Type | Method | JiNan | | | HangZhou | | New York | |
|------|--------|-------|-------|-------|----------|-------|----------|-------|
| | | 1 | 2 | 3 | 1 | 2 | 1 | 2 |
| Online | FRAP | 299.56 ± 1.95 | 268.57 ± 2.03 | 269.20 ± 2.12 | 308.73 ± 1.72 | 355.80 ± 2.32 | 1192.23 ± 11.42 | 1470.51 ± 36.41 |
| | MPLight | 302.36 ± 1.69 | 276.80 ± 1.35 | 267.94 ± 2.69 | 316.41 ± 1.62 | 394.04 ± 18.06 | 1876.8 ± 12.37 | 1642.05 ± 11.23 |
| | CoLight | 272.66 ± 1.84 | 250.95 ± 0.43 | 247.89 ± 0.65 | 295.57 ± 1.61 | 340.24 ± 5.33 | 1078.09 ± 56.81 | 1347.54 ± 40.32 |
| | AttentionLight | 254.82 ± 1.22 | 239.68 ± 1.62 | 236.62 ± 1.92 | 283.64 ± 1.75 | 316.38 ± 2.02 | 1013.78 ± 12.32 | 1401.32 ± 32.32 |
| | Efficient-MPLight | 262.62 ± 1.08 | 240.49 ± 0.38 | 239.54 ± 1.00 | 284.71 ± 0.89 | 321.88 ± 7.91 | 1189.69 ± 53.17 | 1453.52 ± 32.52 |
| | Advanced-MPLight | 258.41 ± 11.87 | 235.22 ± 0.46 | 231.86 ± 0.84 | 273.18 ± 0.54 | 312.74 ± 4.08 | 1498.61 ± 151.28 | 1389.26 ± 51.28 |
| | Advanced-CoLight | **246.41 ± 1.03** | **233.96 ± 0.45** | **229.90 ± 0.69** | **271.51 ± 0.37** | **311.08 ± 4.93** | **924.57 ± 52.02** | **1300.62 ± 25.02** |
| | AttendLight | 277.53 ± 1.69 | 250.29 ± 1.52 | 248.82 ± 1.12 | 293.89 ± 2.05 | 345.72 ± 2.35 | 1586.09 ± 24.23 | 1683.57 ± 36.23 |
| | PRGLight | 291.27 ± 1.62 | 257.52 ± 1.34 | 261.74 ± 1.62 | 301.06 ± 1.33 | 369.98 ± 2.12 | 1283.37 ± 21.23 | 1472.73 ± 42.23 |
| Offline | BC | 282.18 ± 1.86 | 243.52 ± 0.71 | 242.63 ± 0.60 | 364.01 ± 1.76 | 341.82 ± 1.28 | 1132.30 ± 0.00 | 1460.94 ± 0.00 |
| | TD3+BC | 284.70 ± 0.99 | 242.30 ± 0.60 | 258.56 ± 0.60 | 283.31 ± 0.67 | 390.85 ± 5.26 | 1186.08 ± 0.00 | 1536.76 ± 0.00 |
| | BEAR | 277.72 ± 2.93 | 242.43 ± 1.03 | 254.77 ± 2.88 | 288.56 ± 1.76 | 359.62 ± 5.88 | **1033.63 ± 0.00** | 1425.37 ± 0.00 |
| | CQL | 265.68 ± 0.78 | 260.57 ± 0.42 | 234.17 ± 0.38 | 287.31 ± 0.56 | 338.52 ± 1.28 | 1099.83 ± 0.00 | **1405.77 ± 0.00** |
| | Combo | 263.48 ± 0.98 | 257.57 ± 1.52 | **233.17 ± 1.08** | 285.31 ± 1.06 | 348.12 ± 1.78 | 1103.83 ± 0.00 | 1421.57 ± 0.00 |
| | OfflineLight | **263.31 ± 1.16** | **241.13 ± 0.59** | 241.30 ± 0.80 | **281.50 ± 0.93** | **335.33 ± 1.68** | 1076.17 ± 0.00 | 1416.79 ± 0.00 |

Table 1: Performance comparison of different methods (ATT in seconds).

robust stability and performance than other offline RL approaches. Although the CQL performs better in two datasets, BEAR outperforms all methods in one dataset, and Combo shows the best result in one datase, OfflineLight achieves the best performance in four datasets. Furthermore, OfflineLight shows more robust performance among offline-RL methods. More Offline-RL algorithms will be developed based on our Offline-AC framework.

### 6.5 MODEL GENERALIZATION

The OfflineLight has the ability to be directly deployed in the real world after training on the TSC-OID. To better show the capacity of transferability of OfflineLight, we also trained the traditional Advanced-MPLight only on real-world dataset $JiNan_1$, which has shown the best transferability performance Zhang et al. (2022b). Then, the Advanced-MPLight is directly transferred to other real-world datasets after training three epochs to yield the Advanced-MPLight-JN1 (trained on $JiNan_1$). We can see that OfflineLight has shown an obvious advantage of transferability (detailed in Appendix A.1).

### 6.6 MODEL SPEED AND LEARNING CURVE

We also demonstrate that the speed and learning curve of our model have advantages over the baseline methods, and more details can be found in Appendix A.2 and Appendix A.3.

### 6.7 ABLATION STUDY

We remove the constraints of actor and critic in OfflineLight, respectively, for illustrating the importance of these constraints (deails in Appendix C.1).

## 7 CONCLUSION

In this paper, we aim to transform offline data on historical interactions from the environment into robust decision-making systems for traffic signal control (TSC). We propose an abstract method: offline actor-critic and OfflineLight based on it. Our approach is more suitable to be applied in practice without an online interaction with the traffic environment for its decent transferability. Moreover, we collect, organize and release the first TSC offline dataset (TSC-OID) based on multiple real-world traffic flow datasets. The experimental results demonstrate that our approach outperforms most existing methods. Our main contribution is to propose a general offline RL framework (Offline-AC) and prove it generalizable and suitable for developing new offline RL algorithms. Although TSC-OID is generated from a simulator, it can promote the application and development of offline RL for TSC. Moreover, TSC-OID is a bridge connecting to real-world datasets. In the future, we will continue to optimize our OfflineLight model by considering Bayes' theorem for the data distribution issues.

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
