# OpenReview forum: "OfflineLight: An Offline Reinforcement Learning Model for Traffic Signal Control"
_ICLR.cc/2024/Conference — Submitted to ICLR 2024_

### Official Review · Reviewer_aRRe · 2023-10-14

**Soundness:** 2 fair
**Presentation:** 2 fair
**Contribution:** 2 fair
**Rating:** 5
**Confidence:** 4

**Summary:**

RL methods have gained popularity in the traffic signal control (TSC) problem, since they can learn from the traffic environment’s feedback and adjust the traffic signal policies accordingly.
To avoid trial and error training, this paper considers offline RL for adaptive and efficient TSC.
To this end, this paper introduces a general offline actor-critic framework  (Offline-AC)  and develop an adaptive decision-making model, namely OfflineLight, based on the Offline-AC for TSC.
The proposed method improves the generalization ability under a variety of traffic conditions.
The authors also release the first offline dataset for TSC.

**Strengths:**

1. This paper collect and release the first offline dataset for TSC problem, which should benefit the community.
2.  The proposed method shows reasonable performance in the experiments over several datasets.

**Weaknesses:**

1. Several math formulas are incorrect. For example, Eq. 1 should be something like $G = E_{\rho_{0}} E_{a_0, a_1,\ldots, \sim \pi}[\sum_{t=0}^\infty \gamma^t r_t]$, where $\rho_0$ is the initial state distribution. Also problematic is the definition of $\pi^*$ two lines after Eq. 2 and the $Q(s,a)$ in Section 4.2.2 (missing the expectation and what is $r^n$?).
2. The proposed "offline Actor-Critic framework" may not be considered as the contribution of this paper. For example, Eq. 3 in this paper is the same as Eq. 7 in BRAC [1]. Eq. 5 is the same as Eq. 2 in CQL [2].
3. There are many typos and inaccurate words/punctuation marks/notations in the paper, making the paper hard to follow.
4. In Table 1, CQL has similar overall performance but shorter error bars than the proposed method OfflineLight, so it may be overclaim to say that the proposed method "shows more robust performance among offline-RL methods."
5. The main paper is not self-contained, the authors may remove/reduce Section 5 and move the results of Section 6.6 & 6.7 onto the main paper.


[1] Wu, Yifan, George Tucker, and Ofir Nachum. "Behavior regularized offline reinforcement learning." arXiv preprint arXiv:1911.11361 (2019).

[2] Kumar, Aviral, et al. "Conservative q-learning for offline reinforcement learning." Advances in Neural Information Processing Systems 33 (2020): 1179-1191.

**Questions:**

1. What is the reward function you use for the RL formulation of the TSC problem?
2. What this the meaning of this sentence: "the Critic is applied to address overestimated values of the Critic"?
3. I don't understand this sentence in Section 4.3.1: "where $\pi_{\theta'}$ is the target policy network by constraining the policy." Wouldn't the "target policy network" being an exponential moving average of the learning policy as in BEAR [3]?
4. For the sentence "In general, the Actor implementation applies PSR to overfitting to narrow peaks in the value estimate..." Why applying PSR to **overfitting** to narrow peaks?
5. "Our method provides new insights into the OOD problem of offline RL" --- Could you be more specific about what are the new insights?
6. How many seeds do you use in composing Table 1?

[3] Kumar, Aviral, et al. "Stabilizing off-policy q-learning via bootstrapping error reduction." Advances in Neural Information Processing Systems 32 (2019).

---

> ### Author Response · Authors · 2023-11-19
> **Response to viewer aRRe**
>
> The authors express gratitude for the valuable comments and will address them individually.
>
> - Q1：What is the reward function?
>
>   **Response:** The reward function in our decentralized MARL approach focuses on maximizing individual rewards for each traffic signal controller (TSC), as detailed in Equation 2. The metric used is the vehicle queue length (QL), with shorter queues indicating higher efficiency and rewards.
>
> - Q2: What is the meaning of "the Critic is applied to address overestimated values of the Critic"?
>
>   **Response:** Apologies for the misunderstanding due to the inappropriate terminology used. The correct term should be 'overestimated values of the Q value,' not 'overestimated values of the Critic.' We have corrected for this in the revision.
>
> - Q3: In 4.3.1: " where  $\pi_{\theta'}$  is the target policy network by constraining the policy."
>
>   **Response:** Apologies for the lack of clarity in our description within Section 4.3.1 of the 'BEAR' paper, particularly regarding the target policy network. To clarify, the implementation of the target policy network in our paper is indeed similar to that in BEAR. We integrate noise into the learning policy to expand the range of action selection.  Specifically, we implement this by calculating the KL divergence. We will clarify in the revision when the paper is accepted.
>
> - Q4: Why applying PSR to overfitting to narrow peaks?
>
>   **Response:** Firstly, the primary function of PSR in our Actor implementation is not to directly address overfitting to narrow peaks in the value estimate, but rather to enhance the policy's robustness and generalization. The reference to ‘narrow peaks’ is in the context of the value function landscape. In some scenarios, the value function might exhibit sharp peaks, indicating a high estimated reward for very specific actions or action sequences. Without appropriate regularization, the Actor might overfit to these narrow peaks, leading to a brittle policy that performs well in the training environment but fails to generalize to slightly different or unseen situations.
>
> - Q5:  What are the new insights?
>
>   **Response:** Our method introduces a unique computational paradigm that simultaneously considers constraints on policy (action) and value, focusing on the trade-off between these two aspects. This approach is particularly insightful for addressing the OOD challenges in offline RL. By balancing the constraints on policy and value, our method effectively mitigates the common pitfalls associated with distributional shifts, which are prevalent in offline RL scenarios. OfflineLight, under this paradigm, has set a new benchmark in traffic signal control (TSC) within offline RL.
>
> - Q6:  How many seeds do you use in composing Table 1?
>
>   **Response:** To ensure a fair and consistent comparison across all experimental runs, we employed the same seed for each iteration of our experiments. Specifically, we conducted a total of 80 rounds of training, with each round followed by a testing phase. Additionally, our code has been made publicly available to facilitate transparency and reproducibility.
>
> - W1: Several math formulas are incorrect.
>
>   **Response:** We apologize for our carelessness and thank you for your attention. There are several typos in the formula. In Eq.1, $\gamma^{n}$ should be $\gamma^{t}$,
>   and $r^{n}$ of $Q(s,a)$ in Section 4.2.2 should be $r^{t^{'}}$.
>
> - W2: What is the  "offline Actor-Critic framework"'s contribution?
>
>   **Response:** We believe that the main contribution of this paper is the innovation of proposing a new framework. It actually encourages novel approaches and breakthroughs in optimizing and balancing these constraints, which is a significant aspect of our work. Our OfflineLight is an example algorithm for offline Actor-Critic framework, which  lies in integrating and leveraging these known techniques to push the boundaries of performance in TSC.
>
> - W3: The paper is hard to follow.
>
>   **Response**:  We have carefully reviewed and polished our manuscript, making structural adjustments as per your suggestions.
>
> - W4: CQL has similar overall performance.
>
>   **Response:** Our analysis across seven datasets demonstrates that OfflineLight outperforms CQL in five out of these seven datasets. In some cases, the improvements brought by OfflineLight involve significant reductions in Average Travel Time (ATT) by several tens of seconds. While in other datasets, the reduction in ATT may appear marginal (a few seconds), hence it is important to consider the broader impact in urban traffic systems. When translated into real-world implications, these time savings can result in considerable reductions in fuel consumption and subsequent decreases in air pollution.
>
> - W5: The authors may remove/reduce Section 5 and Section 6.
>
>   **Response:** Based on feedback, Sections 5 and 6 will be reduced, with essential results from Sections 6.6 & 6.7 integrated into the main paper for coherence.

---

> > ### Comment · Reviewer_aRRe · 2023-11-21
> > **Responses to the authors**
> >
> > Dear authors,
> >
> > Thank you so much for your reply.
> >
> > After reading other reviewers' reviews and your responses. I do have some follow-up questions.
> >
> > 1. Could you elaborate more on "*the Critic is applied to address overestimated values of the Q value*"? I still cannot understand this statement even after your clarification in the rebuttal. What is your definition of Critic? How is it different from a (conservative/penalized) Q-function?
> >
> > 2. For your statement that "*... leading to a brittle policy that performs well in the training environment but fails to generalize to slightly different or unseen situations,*" this is not the goal of standard RL, which is indeed to maximize performance in the training environment. Could you be more specific on the goal of your RL method? As an aside, there is a huge pool of papers on RL-agent generation, multi-task learning, fast adaptation in the multi-environment setting, and so on, which are not related to this paper and should not be confused with the proposed method.
> >
> > 3. Why do you want to simultaneously considers constraints on policy and value? Why isn't constraining one of them sufficient for offline-RL training as demonstrated in CQL, BRAC, BEAR and so on.
> >
> > 4. About novelty:
> > Could you elaborate more on how is the proposed method different from the papers I listed in the **Weaknesses** section of my previous review?

---

> > > ### Author Response · Authors · 2023-11-22
> > > **Response to aRRe**
> > >
> > > Thank you for your continued engagement, constructive comments, and valuable feedback. We are pleased to note that our previous rebuttal has clarified most of your concerns. Your insights are invaluable in guiding us to provide more detailed explanations and enhance our work's overall quality.
> > >
> > > - **Q1'**:  Elaborate more on "the Critic"?
> > >
> > > **Reponse**:
> > >
> > > 1. In our work, the Critic within the OfflineLight framework serves as a critical component of the Offline-AC framework. The Critic's role is to evaluate the policy's performance by estimating the conservative Q-values.
> > >
> > > 2. The constraint applied to the Critic's Q-value in OfflineLight mirrors the constraint condition in CQL. However, as we theoretically prove it in Appendix D, this approach allows both the Actor and Critic in OfflineLight to achieve more stable results in offline RL scenarios.
> > >
> > > 3. The innovation herein lies in the simultaneous constraint of both the Actor and Critic, a methodology that has not been explored before in this context of domain. This dual-constraint approach not only adheres to the Offline-AC computation paradigm but also opens avenues for further innovations within a bigger framework, where we have already carried out extensive researches and real-world experiments in the TSC problem.
> > >
> > > - **Q2'** : Provide more specific on the goal of your RL method.
> > >
> > > **Response** :
> > >
> > > 1.  Our primary contribution lies in the generalization capabilities. Trained on only 20% of the data, it can be effectively deployed in different real traffic topologies and flows. Even if the New York's topology and flows have not been collected, we can directly deploy to its scenario. Because the OfflineLight is decentralized MARL with individual rewards, the difference of traffic topology has little influence on the final performance.
> > >
> > > 2. The objective of our RL approach is to conduct decentralized training on a dataset representing diverse traffic conditions, with each agent focusing on minimizing queue length (QL).  As detailed in Section 3.3. A shorter queue length signifies more efficient traffic flow and a higher reward.
> > >
> > > 3.  While extensive researches on RL-agent exist, these studies predominantly focus on domains like robotics and gaming, where scenarios are relatively fixed and can be exhaustively trained on. However, the TSC task presents unique challenges due to the variability in intersection topologies and traffic volumes, making RL generalization more complex.
> > >
> > > - **Q3'** : Why does constraint on policy and value?
> > >
> > > **Response**:
> > >
> > > 1. The classical BEAR method only constrained policy while the CQL method only constrained the value. Therefore, we wanted to design a computational framework that constrains policy and value simultaneously. BEAR and CQL are just specific implementations of this general framework. We believe we have established a unified model foundation for offline RL.
> > >
> > > 2. The OfflineLight method simply constrains policy in the same way as BEAR while constraining value in the same way as CQL, achieving a new SOTA for offline RL on TSC tasks, as reported in our paper and appendices.
> > >
> > > 3. We have theoretically proven that OfflineLight satisfies the requirements for offline RL, and we hope this spurs more Offline RL methods to be proposed within this paradigm. We believe simultaneously constraining policy and value is a key to further advancements in offline RL.
> > >
> > > - **Q4'**: About novelty.
> > >
> > > **Response**:
> > >
> > > 1.  BRAC is a framework focusing on behavior regularization through BEAR, BCQ, and KL-Control; it primarily centers around policy-based methods. In contrast, our Offline-AC framework uniquely combines policy and value constraints. This unification broadens the scope from a purely policy-based perspective to include value-based approaches. Specifically, OfflineLight employs BEAR for policy constraints and CQL for value constraints to enhance performance.
> > >
> > > 2.  CQL lies in learning a conservative Q-function but it does not integrate policy constraints. OfflineLight transcends this limitation by incorporating both policy constraints ( PSR in BEAR) and value constraints (conservative Q-value). This dual approach not only addresses the shortcomings of CQL but also expands its utility in offline RL.
> > >
> > > 3.  Our OfflineLight integrates established concepts and SOTA results in offline RL. This achievement itself signifies novelty. In parallel, methods such as TD3+BC [1], which combines the existing TD3 and BC approaches, have led to surprising and noteworthy results. Similarly, OfflineLight's innovation lies in integrating and leveraging these known techniques to push the boundaries of performance in TSC. Moreover, our method has been trialled and evidenced in real-life implementation at some cities, where the outcomes even outperformed the results reported in this paper. Anyhow, we will keep this paper intact for now and share those new results in future publication(s).
> > >
> > > [1] A Minimalist Approach to Offline Reinforcement Learning. NIPS (2021).

---

> > > > ### Comment · Reviewer_aRRe · 2023-11-23
> > > > **Response to authors**
> > > >
> > > > Dear authors,
> > > >
> > > > Thank you so much for the reply. Below are my comments to some of your misunderstandings.
> > > >
> > > > > simultaneously constraining policy and value...
> > > >
> > > > As far as I know, it can be redundant to simultaneously constraint both policy and value function(s). For example, as suggested in the CQL paper, if we constraint the Q-function, then the policy learning can be left unconstrained and thereby optimized towards a better optimal (guided by the conservative Q-function).
> > > >
> > > > >  CQL ... does not integrate policy constraints. OfflineLight transcends this limitation...
> > > >
> > > > As far as I know, for policy optimization, no constraint should be a advantage rather than a disadvantage.
> > > >
> > > > > In parallel, methods such as TD3+BC [1], which combines the existing TD3 and BC approaches, have led to surprising and noteworthy results.
> > > >
> > > > I agree. But the title of the TD3+BC paper is "*A **Minimalist** Approach to Offline Reinforcement Learning*", which is different from your proposal and contribution. Therefore, I don't think your paper is in parallel to TD3+BC, and the comparison with it may not be fair.
> > > >
> > > > ***
> > > > Nevertheless, I agree that the empirical result of this paper has its own merit. I have increased my rating to 5.

---

> > > > > ### Author Response · Authors · 2023-11-23
> > > > > **Thank you for your comments**
> > > > >
> > > > > Thank you for your detailed feedback. We appreciate the opportunity to clarify these points further.
> > > > >
> > > > > We agree with your observation that in offline RL, it is just sometimes necessary to constrain the policy and value functions simultaneously. Indeed, as CQL demonstrated, constraining the value function only can be effective, as policy learning is optimized and guided by the conservative Q-function. In our work with OfflineLight, we have incorporated this understanding into the Offline-AC framework, which accommodates both policy-only constraints (as in BEAR) and value-only constraints (as in CQL).
> > > > >
> > > > > We have provided both theoretical and experimental evidence demonstrating the advantages of OfflineLight over the existing methods like CQL and BEAR. Our approach integrates the strengths of both policy constraints (akin to PSR in BEAR) and value constraints (similar to conservative Q-value in CQL), aiming to achieve a more robust and effective solution in the offline RL context.
> > > > >
> > > > > While acknowledging the difference between the approach and the title of the TD3+BC paper, we believe the comparison is still pertinent. Our reference to TD3+BC was to illustrate how the integration of the existing methods (TD3 and BC) can lead to significant advancements. Similarly, OfflineLight combines and innovates upon the established techniques (policy constraints from BEAR and value constraints from CQL) to enhance performance in traffic signal control.
> > > > >
> > > > > We understand that our approach differs from TD3+BC, but the underlying principle of integrating and optimizing the existing methods to outperform SOTA is a shared aspect of both our works.
> > > > >
> > > > > Given the theoretical and practical advancements demonstrated by OfflineLight and its potential impact on enhancing efficiency and reducing emissions in intelligent transportation systems, we respectfully hope you could further consider to raise the score of our paper for potential publication. We are committed to contributing valuable insights to the ICLR community and the broader RL and the smart transportation field.
> > > > >
> > > > > Again, Thank you for your constructive feedback and further consideration of raising our score.

---

### Official Review · Reviewer_46VN · 2023-10-29

**Soundness:** 3 good
**Presentation:** 3 good
**Contribution:** 2 fair
**Rating:** 6
**Confidence:** 5

**Summary:**

This paper creates an offline dataset (TSC-OID) collected with trained reinforcement learning policies on real-life datasets for traffic signal control task (TSC) to stress the problem that the trial and error training procedure is unsuitable for the traffic signal control problem. Based on the collected dataset, the author also proposed a novel offline-AC algorithm to train offline agents, taking a conservative exploration strategy. The result shows that the offline trained agents have performance close to SOTA online training agents and, at the same time, have the ability to transfer to unseen datasets.

**Strengths:**

1. The motivation for this work is good. TSC is not an environment like a based environment; the traditional trial and error exploration strategy is unsuitable for this high-stack training strategy. The idea of taking online training of TSC to offline training is necessary and important
2. The paper is mostly well-written, with a clear description of the motivation and algorithm. Most of the motivation for designing the offline-AC is clearly explained.
3. The experiment result is consistent with the claims made in the paper, and the transferability discussion is essential to justify the motivation, which is why we need to train an agent offline with a dataset collected from trained RL agents.

**Weaknesses:**

1. The novelty is not clear for this work. The offline-AC brings in ideas from policy smooth regularization (PSR), and conservative Q-learning combines these ideas and transfers them into the application of traffic signal control. Though the attempts are reasonable, the novelty is not recognized as a novelty closely related to representation learning.
2. Though the algorithm is well explained, the most crucial part is that the dataset is not well explained. Some details are very critical to judge the quality of this dataset. For example, how many policies did the author use to collect the data? Does this policy train on all datasets? These are the major components of this paper but need to be well discussed.

**Questions:**

1. For the dataset, how did the author collect data? Is the policy trained on some other dataset or all the datasets used later? Is the RL agent already well-trained before interacting with the environment? If it is trained on all datasets, then the transferability evaluation is not feasible to evaluate the generalization of this offline-AC performance. Could the author give a very detailed description of this part?
2. In the appendix Figure 4. The comparison seems not convincing. If the advanced MPLight performs very well on the transferred dataset, then the transfer ratio is also low. Could the author provide another metric to evaluate the performance?
3. In section 4.2.1, the J(\theta) is calculated over n and t. But in the formula, Q is independent of n and t. Should the action and state subscribe with n and t?
4. How could the author conclude the Discussion “Why is the Offline-AC the abstract method?”. To justify this idea, more experiment results should be conducted.

---

> ### Author Response · Authors · 2023-11-19
> **Response to viewer 46VN**
>
> We thank you for your insightful reviews and address your concerns as follows.
>
> - Q1&W2：Describe Offline Dataset Collection and Training Procedure?
>
>   **Response:**
>
>   1.Data Collection Method: Our approach to data collection prioritizes diversity to enhance the model's generalization capabilities. We used a variety of states and rewards, acknowledging that the optimal choices in existing works are still ambiguous. Our offline dataset comprises interactions generated by different policies, akin to the data buffers in off-policy reinforcement learning methods like Q-learning or DDPG. We engage various RL methods to interact with the CityFlow traffic simulator, recording diverse interaction data. This process encompasses different frameworks, traffic states, and rewards, detailed in Table 3 in Appendix A.2.
>
>   2.States of RL Agents: The RL agents employed in our study were trained from scratch. This approach ensures that the distribution of rewards does not converge prematurely, allowing us to discern the impacts of different actions more effectively. If agents had been well trained beforehand, the reward distribution would have stabilized, making it difficult to assess the relative merits of their actions.
>
>   3.Training on Datasets: Our training was conducted on a limited portion of the available datasets. For example, OfflineLight-JN1 was trained on only 20\% of all data from TSC-OID. Despite this limited training, our model demonstrates robustness, as it can be effectively deployed on various other traffic topologies and flows without extensive retraining.
>
> - Q2:In the appendix Figure 4. The comparison seems not convincing.
>
>   **Response:** It is essential to clarify the interpretation of the transfer ratio used in our study. The transfer ratio, defined as $t_{transfer}/t_{train}$ , inversely correlates with performance: The higher the ratio, the lower the performance.
>   The MPLight  has shown the best transferability performance. But the comparison demonstrates that advanced MPLight does not outperform OfflineLight in Figure 4. The seemingly low transfer ratio for MPLight does not indicate superior performance but rather the opposite. This is a crucial aspect of our analysis, highlighting the effectiveness of OfflineLight in comparison to MPLight under the given metric.
>
>   To further solidify our argument about the generalization capabilities of OfflineLight, we plan to include several classic RL methods in our analysis.
>
> - Q3:In section 4.2.1, should the action and state subscribe with n and t?
>
>   **Response:** It is crucial to note that the Q value in our formulation indeed corresponds to each time step t in the $n^{\text{th}}$ batch. While in Equation (3), for the sake of simplicity, the Q value is not explicitly subscripted with n and t , the calculations preceding this formulation have taken these variables into account. The decision to present the Q value without direct reference to n and t in the equation was a deliberate choice aimed at simplifying the representation and making the equation more readable. However, this should not be misconstrued as an oversight or a simplification in the actual computational process. We will certainly clarify this in the revision.
>
> - Q4:“Why is the Offline-AC the abstract method?”.
>
>   **Response:**
>
>   1.Offline-AC (Actor-Critic) should be recognized as a computational paradigm within offline RL, uniquely integrating constraints on actions and values. This paradigm is not merely a specific algorithm or technique but represents a broader conceptual approach.
>
>   2.Methods like CQL and BEAR can be viewed as specific instances of the Offline-AC framework. OfflineLight is also implemented under this paradigm. These examples demonstrate the versatility and applicability of the Offline-AC approach in various contexts.
>
> - W1:The novelty is not clear for this work.
>
>   **Response:**
>
>   1. Innovation of Offline-AC: Offline-AC is a computational paradigm within offline RL that emphasizes innovation in the constraints on actions and values. This framework is not only applying existing methods; instead, it actually encourages novel approaches and breakthroughs in optimizing and balancing these constraints, which is a significant aspect of our work.
>
>   2. Innovation in OfflineLight:Our model, OfflineLight, utilizing established concepts such as PSR and Conservative Q values, achieves SOTA results in offline RL, yielding impressive outcomes. This achievement itself signifies novelty. In parallel, methods such as TD3+BC [1], which combines the existing TD3 and BC approaches, have led to surprising and noteworthy results. Similarly, OfflineLight's innovation lies in integrating and leveraging these known techniques to push the boundaries of performance in TSC.  Moreover, our method can be evidenced in our real implementation at some cities.
>
>   [1] A Minimalist Approach to Offline Reinforcement Learning.

---

> > ### Comment · Reviewer_46VN · 2023-11-21
> >
> > Thanks for your response. My concerns are addressed by the rebuttal and I now raise my score. My remaining suggestion is to differentiate the novelty of this work in the paper by including your response to my W1 in the paper.

---

> > > ### Author Response · Authors · 2023-11-21
> > > **Response to Reviewer 46VN**
> > >
> > > Thank you for your recent feedback and raising your score. We are pleased to hear that our responses have successfully addressed your concerns.
> > >
> > > We completely agree with your valuable suggestion to emphasize the novelty of our work in the paper. We will incorporate a detailed explanation, similar to our response to your W1 query, to delineate our research's unique aspects and contributions. This addition will significantly enhance the clarity and impact of our paper.

---

### Official Review · Reviewer_Twe3 · 2023-10-31

**Soundness:** 2 fair
**Presentation:** 2 fair
**Contribution:** 2 fair
**Rating:** 5
**Confidence:** 2

**Summary:**

This paper introduces Offline-AC, a general offline actor-critic framework for traffic signal control, addressing the limitations of traditional RL methods that rely on costly trial and error training. They also propose OfflineLight, an adaptive decision-making model based on Offline-AC. Additionally, the paper presents TSC-OID, the first offline dataset for traffic signal control, generated from state-of-the-art RL models, and demonstrates through real-world experiments that Offline RL, especially OfflineLight, can achieve high performance without online interactions with the traffic environment and offers impressive generalization after training on only 20% of the TSC-OID dataset.

**Strengths:**

It's evident that this paper addresses a significant challenge in the field of RL-based traffic signal control by focusing on training policies using offline datasets. This approach is highly practical, as acquiring online samples from high-fidelity traffic simulators like CityFlow and SUMO can be challenging, particularly in scenarios involving large and dense traffic networks. The fact that the proposed offline dataset is publicly accessible is a commendable aspect, as it not only supports the research presented in the paper but also encourages and facilitates further studies and advancements in this area.

**Weaknesses:**

To the best of my understanding, OfflineLight treats each traffic signal as an independent RL agent. Nevertheless, since there are multiple agents, it is crucial to clarify the definitions of state, action, and reward in order to comprehend the problem thoroughly. It would be highly beneficial if there were a well-defined problem formulation, possibly following a POMDP framework. Additionally, I'm interested in gaining a clearer understanding of the objective function. It seems unusual to aim for maximizing the expected reward over historical trajectories when the rewards for these trajectories are already given. Both Offline RL and Online RL generally share a common objective, which is to find an optimal policy that maximizes the expected return within the true MDP. I would appreciate a more detailed elaboration on your objective.

It's worth noting that OfflineLight does not take into account the interactions between multiple traffic lights. In contrast, many studies in Traffic Signal Control (TSC) have explored such interactions using various techniques like the CTDE framework and graph neural networks. I would like to suggest that integrating these approaches could substantially enhance the performance of the proposed model.

Regarding offline RL, it's important to emphasize the significance of the size and quality of the offline dataset for the algorithm's performance. Unfortunately, there is a lack of analysis regarding the dataset's quality. Providing statistics on the offline dataset, such as the maximum reward contained within it, would greatly enhance the understanding of its characteristics.

**Questions:**

I have some detailed questions regarding your work:

1. State and Reward Definitions: Could you please provide more information about the definition of the state? Is the state considered a global state or local information for each agent? Additionally, is the reward shared among all agents or individually assigned? This is a critical matter to address, as most RL-based Traffic Signal Control methods operate in a decentralized Multi-Agent Reinforcement Learning (MARL) framework with a Partially Observable Markov Decision Process (POMDP) setting.

2. Offline Dataset Collection Procedure: I'm interested in understanding the specifics of the offline dataset collection procedure. According to appendix B.2, the offline dataset is collected through three epochs of training. However, this may seem insufficient to attain high-reward solutions. Furthermore, I couldn't find information about which RL method is used to generate the dataset for each scenario. Lastly, could you provide some statistics regarding the reward distribution in the dataset, as the quality of the dataset is crucial for Offline RL performance.

3. Experiment Section: Can you provide more details about the training and evaluation procedures in your experiments? I'm particularly curious about how the offline RL models are trained and evaluated in the New York scenario, given that there is no available offline dataset. Please elaborate on this aspect.

I hope you can provide more insight into these questions to better understand your work.

---

> ### Author Response · Authors · 2023-11-19
> **Response to viewer Twe3**
>
> Thank you for the insightful comments. We herein address them one by one.
>
> - Q1&W1：Definitions of State and Reward.
>
>   **Response:**
>
>   We appreciate your suggestion regarding the need of more detailed information about the state definition in our methodology. Our approach is grounded in a decentralized MARL framework, operating within the confines of a POMDP. This is an integral aspect of our research design. In Section 3.3 of our paper, we provided a comprehensive formulation that delineates the essential elements of our approach, particularly as they pertain to the POMDP context. This section is crafted to offer clarity and depth to our methodological framework.
>
>   Each agent in our model concentrates on local traffic information regarding state representation. This includes NV,  NV-segments, QL, EP,  ERV, and TMP for training process.
>
>   All these state variables are integrated into our OfflineLight model. However, in practical applications, we primarily utilize the queue length (QL) as the state input. This decision is based on our assessment of QL's relevance and effectiveness in capturing the traffic state's critical aspects for our model's objectives. This can be evidenced in our real implementation at some cities.
>
>
>
> - Q2&W3:Describe Offline Dataset Collection Procedure.
>
>   **Response:**  We appreciate your comment and would like to provide clarifications as follows:
>
>   1. Only Three Epochs of Training:
>      (1) Our dataset is unique as it incorporates real-world traffic topology and flow data. This approach ensures that each epoch encompasses a significantly large volume of data. The substantial data volume per epoch compensates for what may initially appear as a limited number of epochs.
>      (2) The initial training epochs are particularly crucial for our analysis. In these stages, the reward distribution exhibits a wide range of variability, which is fundamental for algorithmic guidance. This concept is somewhat similar to the Advantage mechanism in Actor-Critic methods like A2C, where the difference between Q-values and value functions is most pronounced in the early stages of learning.
>
>   2. RL Methods Used: To address your query regarding the specific RL methods used for dataset generation in each scenario, we would direct your attention to Appendix Table 5 of our documentation.
>
>   3. Reward Distribution: We show the distribution of reward of our offline dataset in an anonymous URL : https://anonymous.4open.science/r/OfflineLight_rebuttal-3BA9/README.md.
>
>
>
> - Q3:Provide more details about the training and evaluation procedures in your experiments?
>
>   **Response:** We actually detailed the OfflineLight training process in Appendix A. It can be found that OfflineLight quickly converges in the first three training episodes and shows exceptionally stable and uniformly smooth learning curves across five real-world datasets. One of our main contributions is the generalization of the model. Although OfflineLight-JN1 trained in 20\% of all data from TSC-OID, it can be deployed directly on other traffic topologies and flows. Intuitively pleasing but with little surprise, OfflineLight-JN1 has almost achieved a similar performance to the SOTA RL methods. Even if the New York's topology and flows have not been collected to generate the offline dataset, we can directly deploy to its scenario.  Because the OfflineLight is decentralized MARL with individually reward, the difference of traffic topology has little influence on the final performance.
>
> - W2:OfflineLight does not take into account the interactions between multiple traffic lights.
>
>   **Response:**
>
>   We acknowledge the potential benefits of integrating approaches that consider interactions between multiple agents.  In our next phase of development, we aim to explore the incorporation of offline multi-agent interactions to further improve the results in TSC. However, the current OfflineLight has the advantages as follows:
>
>    1. Our OfflineLight has shown superior performance in offline RL scenarios. Notably, it outperforms methods like CoLight and AttendLight (baseline methods in this paper), which do take into account the interactions between various intersections. This outcome indicates that, in certain contexts, the decentralized approach of OfflineLight can be more effective than interaction-centric models. Moreover, OfflineLight is decentralized MARL with individual reward, and the difference of traffic topology has little influence on the final performance.
>
>   2. Comparison with CTDE Methods: We have conducted comparative analyses with methods such as CTDE. Interestingly, OfflineLight exhibited better performance even in these comparisons. MPLight and CoLight are typical CTDE methods, but our OfflineLight performs better. Additionally, a significant advantage of OfflineLight is that it does not require interaction with the environment, which can be a crucial factor in real-world applicability.

---

> > ### Comment · Reviewer_Twe3 · 2023-11-23
> > **Thank you for your response.**
> >
> > Although the rebuttal has addressed my concerns, I still believe this paper lacks novelty and applicability in real problem setting. I have raised my score to 5.

---

> > > ### Author Response · Authors · 2023-11-23
> > > **Thank you for your comments**
> > >
> > > We really appreciate the chance to disseminate our excellent results at ICLR though our original version lack a bit of clarity. We will make proper revision according to your comments to highlight the novel contribution and current great empirical outcome.
> > >
> > > We understand your concerns regarding the novelty and applicability of our paper. However, we would like to emphasize that the innovation of OfflineLight lies in the unique integration and application of the established concepts like PSR (Predictive State Representations) and Conservative Q values. The proposed approach has enabled us to achieve state-of-the-art (SOTA) results in offline reinforcement learning, particularly in the challenging problem of Traffic Signal Control (TSC) in transportation field.
> > >
> > > The novelty of our work is akin to that demonstrated in methods such as TD3+BC [1] (Details can be referenced to our relevant publication). TD3+BC, which combines the existing TD3 and BC approaches, has been recognized for its promising and noteworthy results. Similarly, OfflineLight's innovation is not to create entirely new concepts but to creatively leverage and integrate the well-developed techniques to extend the boundaries of performance in TSC.
> > >
> > > Moreover, we have implemented and validated our methods in practical settings to address real-world applicability with significant results. We plan to further substantiate this claim by publishing detailed results of trials in multiple cities on our GitHub repository to show the performance of OfflineLight. This will provide concrete evidence of the method's effectiveness and real-world applicability.
> > >
> > > We hope this explanation clarifies the novelty and practical value of our work. We are committed to advancing the field of offline RL, and our approach significantly contributes to the academic community and real-world applications. Thus, we hope our response can further clarify your concerns to allow our work to achieve a better score.
> > >
> > > [1] A Minimalist Approach to Offline Reinforcement Learning.

---

### Meta-Review · Area_Chair_kpAo · 2023-12-05

**Metareview:**

This paper proposes a new OfflineLight algorithm which uses offline actor-critic (Offline-AC) to learn traffic signal control while respecting policy and value constraints.  A new offline dataset is also provided from a traffic simulator that is based on real-world road intersections and traffic flow.

The reviewers generally feel that the paper's novelty is limited, and its applicability in real problems needs a better demonstration.  The fairness of comparing to TD3+BC also needs better justification.  Overall, I agree that the paper needs another round of revision before being published.

**Justification For Why Not Higher Score:**

The reviewers generally feel that the paper's novelty is limited, and its applicability in real problems needs a better demonstration.  The fairness of comparing to TD3+BC also needs better justification.  Overall, I agree that the paper needs another round of revision before being published.

**Justification For Why Not Lower Score:**

N/A

---

### Decision · Program_Chairs · 2024-01-16

Reject